# Late Onset of Primary Hemophagocytic Lymphohistiocytosis (HLH) with a Novel Constellation of Compound Heterozygosity Involving Two Missense Variants in the *PRF1* Gene

**DOI:** 10.3390/ijms25052762

**Published:** 2024-02-27

**Authors:** Alina Stadermann, Markus Haar, Armin Riecke, Thomas Mayer, Christian Neumann, Arthur Bauer, Ansgar Schulz, Kumar Nagarathinam, Niklas Gebauer, Svea Böhm, Miriam Groß, Michael Grunert, Matthias Müller, Hanno Witte

**Affiliations:** 1Department of Hematology and Oncology, Bundeswehrkrankemhaus Ulm, Oberer Eselsberg 40, 89081 Ulm, Germany; alinastadermann@bundeswehr.org (A.S.); arminriecke@bundeswehr.org (A.R.);; 2Department of Intensive Care Medicine, University Medical Center Hamburg-Eppendorf, Martinistraße 52, 20251 Hamburg, Germany; 3Department of Pediatric Medicine, University Hospital Ulm, Eythstraße 24, 89075 Ulm, Germany; 4Institute of Biochemistry, University of Lübeck, Ratzeburger Allee 160, 23538 Lübeck, Germany; 5Department of Hematology and Oncology, University Hospital Schleswig-Holstein Campus Lübeck, Ratzeburger Allee 160, 23538 Lübeck, Germany; 6Institute for Immunodeficiency, Medical Center-University of Freiburg, Faculty of Medicine, University of Freiburg, Breisacher Straße 115, 79106 Freiburg, Germany; 7Division of Pediatric Stem Cell Transplantation and Immunology, Department of Pediatric Hematology and Oncology, University Medical Center Hamburg-Eppendorf, Martinistraße 52, 20246 Hamburg, Germany; 8Department of Nuclear Medicine, Bundeswehrkrankenhaus Ulm, Oberer Eselsberg 40, 89081 Ulm, Germany; michael.grunert@uni-ulm.de; 9Department of Nuclear Medicine, University Hospital Ulm, Albert-Einstein-Allee 23, 89081 Ulm, Germany

**Keywords:** primary HLH, PRF1 gene, compound heterozygosity

## Abstract

Hemophagocytic lymphohistiocytosis (HLH) is a rare but in most cases life-threatening immune-mediated disease of the hematopoietic system frequently associated with hematologic neoplasms. Here, we report on a case in which we detected a novel constellation of two missense variants affecting the *PRF1* gene, leading to de novo primary HLH. Diagnostics included a comprehensive clinical work-up and standard methods of hematopathology as well as extended molecular genomics based on polymerase chain reaction (PCR) reactions and the calculation of three-dimensional molecule reconstructions of *PRF1*. Subsequently, a comprehensive review of the literature was performed, which showed that this compound heterozygosity has not been previously described. The patient was a 20-year-old female. Molecular diagnostics revealed two heterozygous missense variants in the *PRF1* gene (A91V and R104C) on exon 2. Apart from the finding of two inconclusive genetic variants, all clinical criteria defined by the HLH study group of Histiocyte Society were met at initial presentation. The final diagnosis was made in cooperation with the Consortium of German HLH—reference centers. Here, chemotherapy did not lead to sufficient sustained disease control. Therefore, the decision for allogenic hematopoietic stem cell transplantation (alloHSCT) was made. Hitherto, the duration of response was 6 months. Due to severe and unmanageable hepatic graft-versus-host disease (GvHD), the patient died. We report on a novel constellation of a compound heterozygosity containing two missense variants on exon 2 of the *PRF1* gene. To the authors’ best knowledge, this is the first presentation of a primary HLH case harboring this genomic constellation with late-onset clinical manifestation.

## 1. Introduction

Hemophagocytic lymphohistiocytosis (HLH) is a rare yet life-threatening disorder characterized by excessive inflammation due to a dysregulated immune state [1]. It is categorized as one of the hyperferritinemic hyperinflammation syndromes within the grouping of histiocytosis [2]. While it is primarily diagnosed in children, with an estimated prevalence of 1 per 100,000, HLH can be triggered at any age by various factors such as infection, malignancy, immunodeficiency, and autoimmune disorders, regardless of the underlying etiology [3]. 

The common pathophysiologic pathway for both genetic ((primary) Familial Hemophagocytic Lymphohistiocytosis 2 (FHL2)) and reactive (secondary) entities of HLH is the inability of natural killer (NK) and CD8^+^ T cells to eliminate antigen-presenting cells (APC) through perforin-dependent and granule-mediated cytotoxicity [4]. This impairment in immune modulation results in the sustained activation of NK, CD8^+^ T cells, and macrophages [5], consequently leading to an excessive secretion of cytokines (cytokine-storm) [6]. This manifests as an acute or subacute febrile condition with multi-organ involvement [3].

Typically, HLH is characterized by the presence of hemophagocytosis in the bone marrow and the lymphatic system, bearing the eponymous hallmark. However, this classical hemophagocytosis manifestation is not universally evident in all patients, posing a challenge for diagnosis [7]. Consequently, diagnostic scoring systems such as the H-score and HLH-2004 criteria are employed to standardize the diagnostic process [8,9]. The clinical presentation often exhibits subtle deviations from that of sepsis. Within the characteristic triad of fever, bi- or pancytopenia, and splenomegaly, a ferritin assessment is recommended when HLH is suspected. In the event of elevated ferritin levels, an expedited consideration of the diverse spectrum of potential causes for secondary HLH is imperative for initiating the fastest possible treatment of the underlying condition. As age advances, the likelihood of HLH attributable to lymphoma rises. A comprehensive diagnostic approach, ideally incorporating Positron-emission tomography (PET)–CT imaging, along with biopsies from the skin, liver, lymphatic tissue, or bone marrow, is advisable. Diagnostic–therapeutic splenectomy may be contemplated in cases of splenomegaly when alternative options are limited. The rigorous diagnostic strategy is justifiable given the markedly dismal prognosis. 

Conversely, primary genetic etiologies must be discerned. In instances where primary HLH is suspected, functional immunological diagnostics, such as the degranulation testing of NK-cells and T-cells, is recommended. Moreover, confirmation of the diagnosis necessitates evidence of an explanatory genetic alteration through mutation analysis. Notably, the proportion of mutation detections not associated with previously identified HLH genes remains substantial, standing at 10%.

Perforin, being one of the key effectors of immune modulation, is coded by the *PRF1* gene (10q22.1). Mutations within this gene are responsible for 20–40% of FHL2 cases and a significant majority of these are homozygous [6,10]. Additional genetic modifications impact the functionality of the inflammasome, specifically involving *NLRC4* and *XIAP*, or cytotoxic exocytosis mediated by *UNC13D*, *STX11*, *STXBP2*, *RAB27A*, and *LYST* [11,12,13]. In the latter instances, a discernible reduction in CD107a expression is observable through flow cytometry (FACS). Furthermore, instances of HLH linked to infrequent genetic constellations, such as alterations in *RAG1 and 2*, *BTK*, *ATM*, *STAT1*, and others, have been documented in the scientific literature [14,15,16,17].

## 2. Methods

### 2.1. Clinicopathological Characteristics

First, the case history was worked up chronologically. Clinical and laboratory characteristics were collected from the original electronic patient file. The diagnosis was made in accordance to clinical criteria defined by the HLH study group of the Histiocyte Society. In this case, HLH diagnosis was implemented in cooperation with the Consortium of German HLH—reference centers. Hematopathologic diagnostics were performed at the Institute of Pathology and Molecular Pathology, Bundeswehrkrankenhaus Ulm and the reference center for haematopathology, University Ulm. This included conventional microscopy and immunohistochemistry.

### 2.2. Immunological Analysis, Molecular Diagnostics and Biochemical Analysis

Functional immunological diagnostics employing perforin expression analysis and degranulation tests for NK- and T-cells were conducted by the Advanced Diagnostics Unit at the Institute for Immunodeficiency, University Medical Center Freiburg, utilizing flow cytometric examinations as published previously [18]. Briefly, peripheral blood mononuclear cells (PBMC) were isolated from whole EDTA blood by standard density gradient centrifugation. To analyze perforin expression in NK and T cells, freshly isolated PBMCs were stained with antibodies against CD3, CD16 and CD56 for 20 min at 4 °C followed by intracellular staining with the anti-perforin antibody for 20 min at 4 °C using the IntraPrep Permeabilization Reagent (Beckman Coulter, Brea, CA, USA).

The subsequent molecular genetic diagnostics were performed by targeted sequencing of the coding exons 2 and 3 of PRF1, including flanking intron boundaries. Primer sequences are available upon request. This analysis was carried out at the molecular biological HLH reference laboratory of the Children’s Hospital at the University Medical Center Hamburg—Eppendorf. To evaluate the pathogenicity of the detected alterations, an analysis was performed with the online prediction program “PolyPhen2” [19].

In addition, we reconstructed the PRF1 molecule in its three-dimensional structure, taking into account the molecularly detected alterations. Plots were generated using PyMol by Schroedinger version 2.5.4. 

### 2.3. Review of the Literature

The methodology for data extraction and analysis entailed a systematic query of the ClinVar database, cBioPortal and PubMed to identify mutations in the *PRF1* gene correlated with Familial Hemophagocytic Lymphohistiocytosis 2 (FHL2) [20,21]. Publications from the period 2000–2023 were considered, which were written in English or German. The search parameters were stringently defined to encompass all variants within the *PRF1* gene, annotated as ‘pathogenic’ or ‘likely pathogenic’, and resulting in an amino acid alteration, ensuring the inclusion of only those genetic aberrations that have functional implications at the protein level and are clinically significant. The dataset was further refined to encompass only those mutations for which publications were available, thereby enhancing the validity of the dataset with empirically substantiated evidence. Subsequent to data retrieval, a deduplication step was employed, thereby removing any redundant entries. The final dataset comprised unique *PRF1* gene mutations, each corroborated by the scientific literature, thereby facilitating a robust genomic analysis pertinent to FHL2 pathophysiology.

### 2.4. Ethical Statement

Written informed consent of the clinical data and scientific use of biopsy material was obtained from the patient. The local ethics committee of the University of Ulm was consulted in advance. Without assigning a reference number, they classified this publication as unobjectionable from an ethical point of view.

## 3. Results

### 3.1. Case Presentation

In November 2021, a 20-year-old female was admitted to an external hospital with fever and pancytopenia. The attempt of a bone marrow biopsy was unsuccessful as the arteria iliaca communis was accidentally injured. The injured vessel was then treated with a stent. The fever had started a few months earlier in August (Figure 1A). From the timepoint of admission to our clinic, episodes of fever and fatigue presented periodically. Blood sampling revealed pancytopenia (Table 1). Sonography assessment detected a hepatosplenomegaly. Bone marrow biopsy did not provide a conclusive diagnosis. There were no features of lymphoma, leukemia, myelodysplastic syndrome, paroxysmal nocturnal hemoglobinuria or other hematologic neoplasms. Magnetic resonance imaging (MRI) showed no evidence of potential tumor lesions. However, Positron-emission tomography (PET) imaging revealed a polytopic increased metabolism in morphologically unremarkable lymph nodes and in the lymphatic tissue of the nasopharynx, liver and spleen, as well as it being osseous and partly also muscular (here in the gastrocnemius muscle), the intensity of the FDG uptake was not sufficiently compatible with a high-grade proliferative neoplasia (Figure 1B). A relevant fraction of potential infectious causes including human immunodeficiency virus (HIV), Epstein–Barr virus (EBV), cytomegalovirus (CMV) and leishmaniosis were ruled out as well. Consecutively, HLH was suspected as blood sampling revealed increased serum levels of ferritin, triglycerides, the soluble IL-2 receptor and a decrease in fibrinogen levels. In the initial course of disease, the highest ferritin value measured was 1032 ng/mL (reference range 22–112 ng/mL) and the highest value measured for the soluble interleukin-2 receptor was 4691 U/mL (reference range 158–623 U/mL). At this timepoint, six out of eight HLH-2004 criteria were fulfilled. 

In its entirety, the diagnostic process posed significant challenges. The case underwent thorough deliberation in the interdisciplinary tumor board for leukemias at the Comprehensive Cancer Center of the University of Ulm (CCCU) and was further discussed in the multi-institutional HLH reference tumor board. Six out of eight HLH-2004 criteria were met; however, hemophagocytosis in the bone marrow was absent, with limited assessibilty of the specimen. Flow cytometry revealed reduced but detectable perforin expression in NK cells. Consequently, an initial course of immunosuppressive therapy was initiated (Figure 2A).

First, the therapy with intravenous immunoglobulins (IVIG) for 4 days, G-CSF and prednisolone was established, resulting in an increase in leukocytes, hemoglobin and thrombocytes. Moreover, LDH, ferritin and triglycerides decreased. This initially resulted in a favorable hematologic remission characterized by substantial reconstitution of the bone marrow. Guided by the reduced perforin expression in NK cells, sequencing of the bone marrow detected a compound heterozygous mutation containing two missense variants on exon 2 of the *PRF1* gene (c.272C>T p.Ala91Val (A91V) and c.310C>T p.Arg104Cys (R104C)), the former being the most prevalent hypomorphic PRF1 mutation, the latter not previously reported in the literature (Figure 2B). The molecular analysis from peripheral blood substantiated the existence of these mutations. This confirmation also affirmed the germline nature of the identified mutations. Due to the inconclusive genetic constellation of the detected compound heterozygosity, we performed an additional calculation of the tertiary structure of PRF1. The structural changes resulting from the presence of both mutations additionally support the diagnosis of primary HLH (Figure 2C–E). This genetic constellation was consistent with the reduced but detectable perforin expression. Remarkable is the late onset of clinical manifestation in this case, which we classify as primary HLH.

Despite responses of limited duration towards immunosuppressive treatment, the HLH relapsed several times (March 2022, August 2022 and September 2022). At the beginning, the disease could still be controlled by means of immunosuppression medication. However, with disease progression, its severity escalated, necessitating more intensified therapeutic interventions. Given the severity of the disease, the option of allogeneic stem cell transplantation was discussed with the patient. This concept promised to be the only curative strategy. In the further course, the patient received alemtuzumab (three cycles, October–November 2022) and etoposide (one cycle, November 2022) as a bridging strategy. Two grade III toxicities, as per the Common Toxicity Criteria (CTCAE), manifested in association with alemtuzumab therapy. Primarily, a severe allergic shock ensued during the initial administration, and secondarily, acute grade III kidney injury transpired during the treatment course.

In December 2022, the patient underwent allogenic hematopoietic stem cell transplantation (allo-HSCT). The allogeneic hematopoietic stem cell transplant (HSCT) was sourced from an HLA-identical unrelated donor (5.5 × 10^6^ CD34^+^ cells). The conditioning regimen comprised thiotepa, treosulfan, and anti-thymocyte globulin (ATG). Graft-versus-host disease (GvHD) prophylaxis was implemented with mycophenolate mofetil (MMF), prednisolone, and ciclosporin. Supportive measures included the administration of romiplostim and darbopoetin alfa. Throughout the course of allogeneic HSCT, diverse medical challenges emerged. These encompassed subacute right frontal cerebral ischemia, the onset of engraftment syndrome characterized by weight gain, dyspnea necessitating oxygen supplementation, and mild diarrhea, as well as occurrences of clostridium difficile infection and central venous catheter (CVC)-associated infection associated with the detection of pseudomonas aeruginosa and klebsiella pneumoniae. Furthermore, a therapy-associated peripheral polyneuropathy progressively manifested during the disease course. In the initial chimerism analysis, the observed ratio was already at 95/100% within the peripheral blood. Subsequently, a chimerism analysis conducted in the bone marrow shortly thereafter revealed a state of the complete remission of hemophagocytic lymphohistiocytosis (HLH), concurrent with full chimerism. A progression-free survival of 6 months was reached until our patient developed severe chronic graft-versus-host disease (GvHD) of the liver presenting with hepatic encephalopathy grade 1–2. Initially, the dose of ciclosporin A was escalated with the addition of a prednisolone boost. Unfortunately, inevitable liver failure occurred, which could not be controlled by escalated immunosuppressive therapy with prednisolone, tacrolimus, etanercept, ruxolitinib, cyclophosphamide and extracorporal photopheresis. A further complicating event was aspergillus pneumonia. Consecutively, the patient died due to severe GvHD. At the time of death, HLH was still in hematologic complete remission. 

In total, the patient underwent intensive care treatment for 75 days. However, 71 of these treatment days, the patient stayed at an intensive care/intermediate care unit previous to allogeneic HSCT.

Due to the genetic constellation of findings in this case, the family was advised to undergo human genetic counseling. In the course of the donor search, the patient’s sister was considered. A comparable genetic constellation could be ruled out in advance.

### 3.2. Review of the Literature

A total of 108 publications were identified in which 38 different mutations in the *PRF1* gene were described (Table 2, Appendix A). The identified alteration at A91V has been previously linked to the onset of primary HLH, specifically when occurring in a homozygous state. Additionally, the missense mutation R104C represents the initial documentation of this alteration in association with primary HLH. The case presented in this study marks the inaugural report of a novel compound heterozygosity in the *PRF1* gene directly associated with the progression of primary HLH, as per the extensive literature search conducted in this study.

## 4. Discussion

In the context of hyperferritinemic inflammation syndromes, Hemophagocytic Lymphohistiocytosis (HLH) represents a severe condition characterized by hemophagocytosis in vital organs like the bone marrow, lymph nodes, and spleen [2,7]. This life-threatening disorder often poses diagnostic challenges due to its clinical resemblance to sepsis. In its hereditary form, the pathogenesis frequently involves the *PRF1* gene which is crucially responsible for cytotoxic granule provision and function, particularly impacting the cytolytic activity of NK cells and CD8^+^ T cells [24]. This impairment leads to uncontrolled T cell proliferation, cytokine storm, and heightened macrophage activation, culminating in a cascade of inflammatory responses.

The autosomal recessive inheritance pattern, notably affecting the *PRF1* gene, predominantly leads to the manifestation of HLH [2]. Structural analysis emphasizes the impact of these mutations on the *PRF1* gene product, encoding a cytolytic protein crucially present in cytotoxic T cells and NK cells [25,26].

Despite initial uncertainties regarding the novel genetic constellation observed in this case, the patient’s clinical presentation corroborated a diagnosis of primary HLH, substantiating the causative link to the identified genetic makeup. This has been discussed with the consortium of German HLH reference centers. 

Notably, the heterozygous A91V variant in the *PRF1* gene is prevalent in approximately 8–9% of the healthy Caucasian population [27]. This is potentially linked to an increased risk of lymphoma development [24]. Using functional analyses, Martinez-Pomar et al. have already shown that the co-occurrence of the A91V variant in the *PRF1* gene and another variant among the *PRF1* gene results in decreased NK cell cytotoxicity in the sense of compound heterozygosity [10]. The genetic constellation presented here can cause fatal outcomes. However, the evaluation of its presence in homozygous form varies significantly in the literature. It has been documented that the A91V alteration, whether in heterozygous or homozygous form, singularly does not manifest as the complete form of HLH [10,28,29]. Additional functional investigations conducted by various research groups have elucidated distinct maturation patterns and a diminished yet measurable expression of the protein. Concurrently, several instances of late-onset HLH have been reported in the literature when coupled with a second mutation in the *PRF1* gene [30,31]. A limited fraction of case reports can be found in the literature, in which the presence of compound heterozygosity of two variants in the *PRF1* gene also led to the development of severe HLH. However, the reported genetic constellations differ from the constellation reported here [28,32]. 

Extensive investigations, such as those by Feng et al. and earlier studies by Ueda et al., underscore the diverse spectrum of mutations along the *PRF1* gene, expanding our understanding of potential alterations linked to primary HLH [33,34]. In cases where a complete loss of function in the *PRF1* gene is identified, allogeneic stem cell transplantation emerges as a therapeutic consideration [13]. However, it is noteworthy that heterozygous variants typically do not induce complete loss of function in this gene. 

Retrospectively, the question must be asked whether etoposide and ciclosporin should have been used earlier in the treatment sequence. The decision to refrain from employing etoposide and ciclosporin in the initial therapeutic intervention was multifaceted. The rationale encompassed the ambiguous constellation of findings arising from the negative degranulation test and the unique genetic compound heterozygosity, not hitherto documented in the scientific literature. Additionally, the patient’s age at the onset of primary HLH was considerably advanced. Furthermore, concerns were raised regarding the potential for pronounced ovarian toxicity. The therapeutic approach would have undoubtedly escalated to incorporate etoposide if a robust remission had not been achieved, with the initial therapeutic regimen involving IVIGs and prednisolone.

We reported on a novel constellation of the compound heterozygosity of two variants in exon 2 (MACPF) of the *PRF1* gene leading to the development of primary HLH. Thus, after a comprehensive review of the literature, this is the first presentation of such a HLH case. Initially, a complete remission was achieved by alloHSCT, but in the course of time, uncontrollable severe GvHD of the liver developed and the patient died. Additional biochemical analyses were a helpful tool for better understanding the influence of mutations on the three-dimensional structure of molecules. This enables the treating physician to better place previously undescribed alterations in the clinical context.

## Figures and Tables

**Figure 1 ijms-25-02762-f001:**
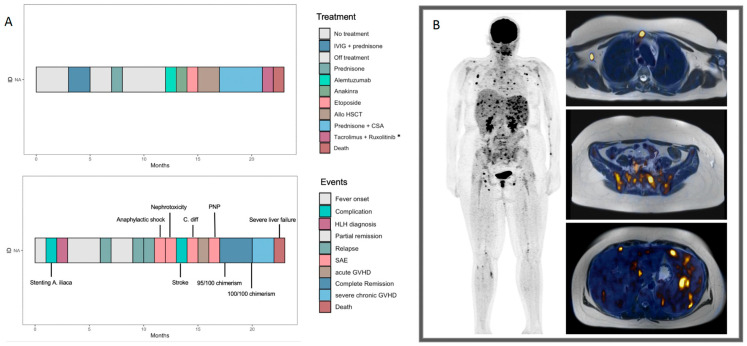
(**A**) The clinical course and treatment sequence. (**B**) Total body ^18^F-FDG PET/MRI in HLH with involvement of several non-enlarged lymph nodes and extranodal organs with increased multifocal FDG uptake, including bone marrow as well as spleen with associated splenomegaly. * During this period, other immunosuppressive agents not listed here in the Swimmer Plot were used (e.g., etanercept).

**Figure 2 ijms-25-02762-f002:**
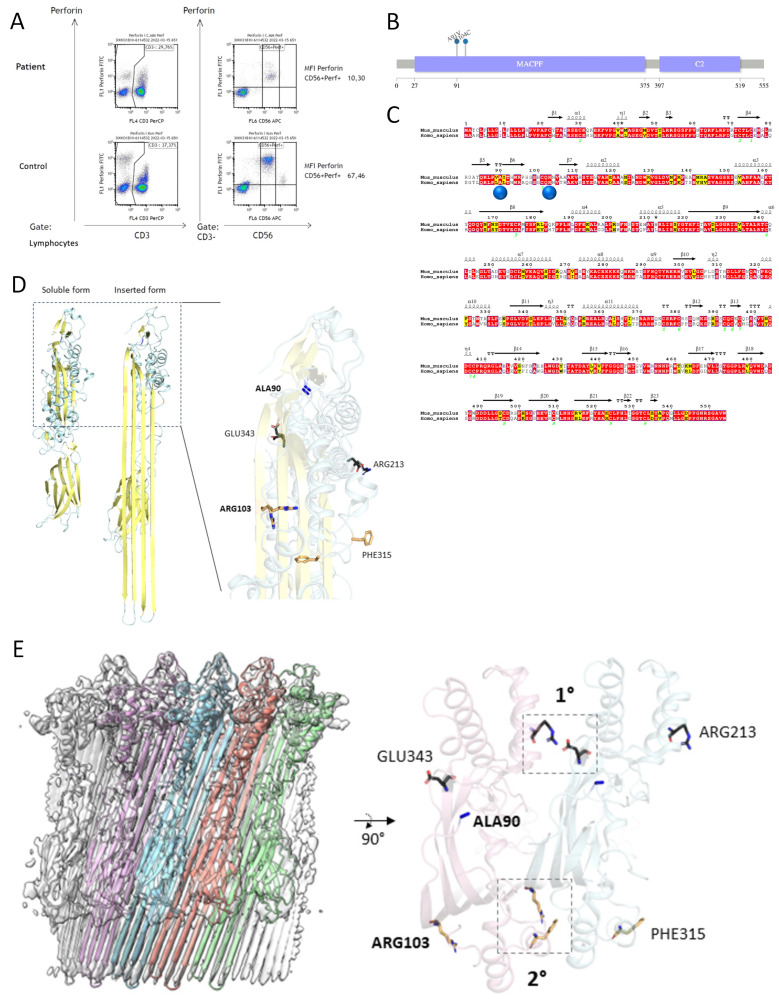
(**A**) The normal perforin expression at mean fluorescence intensity (MFI) was >19.6. The cut-off calculation was performed using ROC analysis on 120 healthy persons and 39 patients. Here, we see a significantly reduced but detectable perforin expression in NK cells, which is consistent with the compound heterozygous mutation in PRF1. (**B**) Lollipop plot of the *PRF1* gene with the alterations detected here in exon 2. (**C**) Alignment of PRF1 gene sequence based on the topology of *Mus musculus* (PDB ID: 3NSJ) with the corresponding mutations (A91V, R104C) in *Homo sapiens* as blue spheres. * The 3NSJ PDB contains multiple rotamers of the sidechains at this position and also ‘T’ stands for turn [22]. (**D**) On the left: cartoon representation of mouse PRF1 in soluble (PDB ID: 3NSJ) and inserted form (PDB ID: 7PAG) showing large conformational changes between the soluble and the extended form (cyan, yellow) of the perforin monomer. On the right: overlay of the soluble and extended form of the perforin monomer highlighted, with the residues involved in the interaction between monomers of the perforin oligomer highlighted as sticks (black, orange), which also includes the mutation ALA90 (blue). (**E**) The Cryo-EM map (EMD-13269) of oligomerized mouse perforin in its inserted state fitted with five monomers of a C22 symmetry pore complex (left inset). Vertical view of two of the monomers (pink and cyan) showing residues also identical in *Homo sapiens* were found to be involved in primary interactions (1°, GLU343, ARG213, black sticks, right inset) at the oligomerization interface, and secondary interactions (2°, ARG103, PHE315, orange sticks) were formed as a result of conformational changes from soluble to extended form [23].

**Table 1 ijms-25-02762-t001:** The course of the laboratory chemical tests.

		2021					2022											
	References	22.11.	02.12.	05.12.	14.12.	27.12.	04.01.	10.01.	18.01.	23.01.	07.02.	21.02.	07.03.	10.03.	20.03.	27.03.	19.05.	15.06.
Leucocytes	4.0–9.0/nL	0.8	0.8	0.8	0.4	5.5	13.7	8.4	1.3	3.2	4.2	3.9	5.1	2.6	4.8	4.7	2.4	3.7
Neutrophiles	1.7–7.2/nL	0.3	0.2	0.2	0.1	3.8	10.4	6.5	0.5	1.5	3.1	3	3.7	2.2	3.6	3.5	1.2	1.2
Hemoglobin	11.5–16.0 g/dL	10	6.5	8.1	7.8	7.6	9.6	8.6	8.5	8.9	11.1	12.2	13.4	11	12.7	12.1	11.1	11.7
Platelets	150–450/nL	28	80	87	101	94	107	88	29	124	92	107	170	46	139	117	159	188
Triglycerides	<1.7 mM				4.3	4.5	6.4						6.6	2.5	3.6			
Ferritin	22–112 μg/L		1032		522	823	3324	3278	577	1751	1568	1169	1317	189	2288	1986	72	86
IL-2-R	158–623 U/mL		4691		4569		2477	1817										
Fibrinogen	1.99–3.43 g/L	2.17	1.7	1.91	1.51	2.08	1.96	1.85	1.63	1.11	2.02	1.97		1.95				2.12
LDH	135–214 U/L	821	694	572	455	242	431	362	424	446	423	477	487	403	451	396	302	265

**Table 2 ijms-25-02762-t002:** Characteristics of the literature review.

Review of the Literature	
Databases Resources	PubMed, cBioPortal, ClinVar
Search terms	PRF1, HLH, hemophagocytosis, hemophagocytic lymphohistiocytosis, familial hemophagocytic lymphohistiocytosis, primary hemophagocytic histiocytosis
Publications	108
Mutations in PRF1 gene	502
Mutations with HLH-phenotype	457
Mutations of clinical significance & impact on amino acid sequence & published	57
Pathogen or likely pathogen	38

## Data Availability

The data presented in this study are available on request from the corresponding author. The data are not publicly available due to compliance with pseudonymization.

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
