# Peer review of "Late Onset of Primary Hemophagocytic Lymphohistiocytosis (HLH) with a Novel Constellation of Compound Heterozygosity Involving Two Missense Variants in the PRF1 Gene"

_ijms, 2024, doi:10.3390/ijms25052762_

Round 1
Reviewer 1 Report
Comments and Suggestions for Authors
Dear Authors
The introduction of the article is very short and the related references that are available have not been used.
(Natalia Martínez-Pomar et al., Functional impact of A91V mutation of the PRF1 perforin gene Hum Immunol. 2013 Jan;74(1):14-7. doi: 10.1016/j.humimm.2012.10.011.
Bruna Maria Bereta de Souza 2022, Perforin and granzyme B gene expression is associated with a short survival time in patients with multiple myeloma. Research, Society and Development, v. 11, n. 14, e111111436237, 2021 (CC BY 4.0) | ISSN 2525-3409 | DOI: http://dx.doi.org/10.33448/rsd-v11i14.36237 1)
The method is written very briefly and references are not provided. The molecular part including enzymes, primers, amplicon size, etc. used in PCR and sequencing should be well presented or the reference used should be added.
The obtained PCR product sequences should be registered in the GenBank and its access number should be presented in the article.
In the discussion section, the used articles are old and new and related references
that indicate the presence of different variants in patients have not been used.
In Figure 2, it is better to draw the prediction of the protein structure for
each amino acid substitutions separately, and its role in oligomerization should be investigated and compared. Comments on the Quality of English Language
-
Author Response
First of all, we would like to thank the reviewer for the numerous helpful comments to improve the manuscript and the assessment of our manuscript. In the following, will now address all comments in a point-by-point fashion.
The introduction of the article is very short and the related references that are available have not been used.
I.(Natalia Martínez-Pomar et al., Functional impact of A91V mutation of the PRF1 perforin gene Hum Immunol. 2013 Jan;74(1):14-7. doi: 10.1016/j.humimm.2012.10.011.
- Bruna Maria Bereta de Souza 2022, Perforin and granzyme B gene expression is associated with a short survival time in patients with multiple myeloma. Research, Society and Development, v. 11, n. 14, e111111436237, 2021 (CC BY 4.0) | ISSN 2525-3409 | DOI: http://dx.doi.org/10.33448/rsd-v11i14.36237 1)
- We thank the reviewer for this comment. Accordingly, we have now presented the introduction in more detail and added the first of the suggested reference (page 4, lines 111 – 112; page 4, lines 123 – 141; pages 4 – 5, lines 144 - 149). As the second reference is not listed in PubMed Central, we have not included it in the list of references. We would like to kindly point out to the reviewer that the first reference mentioned was already cited in the discussion of our manuscript.
The method is written very briefly and references are not provided. The molecular part including enzymes, primers, amplicon size, etc. used in PCR and sequencing should be well presented or the reference used should be added.
The obtained PCR product sequences should be registered in the GenBank and its access number should be presented in the article.
- We agree with the reviewer on this point. The methods section has now also been presented in more detail (page 5, lines 161 – 175). As the diagnostics from the clinical routine were carried out on a service contract, the right to upload the raw data lies with the commercially commissioned laboratory. We requested this in the course of communication about the execution of the methods section. The responsible persons were very reticent to allow an upload.
In the discussion section, the used articles are old and new and related references that indicate the presence of different variants in patients have not been used.
- In order to follow reviewer's request, we once again jointly screened the literature within the working group for more recent references. Consequently, we included more recent literature in the discussion. In particular, the recent publication by Bloch et al. shows the simultaneous presence of different variants in patients with primary HLH. However, and we are now making this even clearer than before, these are cases in which the variants are present in a homozygous constellation (page 8, lines 286 – 288; page 9, lines 311 - 317).
In Figure 2, it is better to draw the prediction of the protein structure for each amino acid substitutions separately, and its role in oligomerization should be investigated and compared.
- We take the reviewer's comment very seriously. We have therefore discussed the illustration again internally. Additional calculations were included and we have now adjusted the figure based on the reviewer's suggestion (page 7, lines 237 – 240; Figure 2C-E).

Reviewer 2 Report
Comments and Suggestions for Authors
In this case report, the Authors described a de novo primary Hemophagocytic lymphohistiocytosis (HLH) patient with a two novel missense variants on PRF1 gene. No molecular insights were given for this genetic alteration, while only a literature review, thus not fullfilling Journal's scopes. Quality of presentation must be improved, as the case is poorly described and figures are in low resolution and not easily readable.
Comments on the Quality of English LanguageMinor checks
Author Response
We would like to thank the reviewer for the appreciative feedback on our work. We have gladly taken on board the points raised and included them in the revised version of the manuscript.
In this case report, the Authors described a de novo primary Hemophagocytic lymphohistiocytosis (HLH) patient with two novel missense variants on PRF1 gene. No molecular insights were given for this genetic alteration, while only a literature review, thus not fullfilling Journal's scopes. Quality of presentation must be improved, as the case is poorly described and figures are in low resolution and not easily readable.
- We apologize for the quality of the images provided. This problem should now be publicly resolved as part of the re-submission. The molecular analyses are now presented in more detail in the methods section (page 5, lines 161 - 175). We also refer to previously published data from our working group in the methods section (Bryceson et al. 2012, Blood). We would like to thank the reviewer for pointing out the adherence to formalities in the manuscript design. After discussing this point internally, we have decided to leave the Review of literature section in the results section as a sub-item, as the content listed here corresponds more to results than discussion points. We hope that this will be comprehensible to the reviewer. Following the reviewer's advice, we have also updated the swimmer plot and changed the color scheme to make it easier to follow (Figure 1A). We have also described the case report in more detail in the revised version of the manuscript (page 6, line 203; pages 6 - 7, lines 217 – 227; page 7, lines 230 – 231; page 7, lines 235 – 242; page 7, lines 244 – 246; page 7, lines 249 – 252; pages 7 – 8, lines 254 – 267; page 8, lines 268 - 279). For the revision, we obtained the expertise of other experts in the field who were involved in the case and included them in the list of authors. The critical evaluation of these experts also led us to adjust the title and make minor additions to the abstract (page 3, Lines 75 and 92). We hope that we have been able to adequately address all the points raised by the reviewer.

Reviewer 3 Report
Comments and Suggestions for Authors
Stadermann et al. present a case report of a young patient with Hemophagocytic syndrome which presents two PRF1 mutations not previously described in the literature.
I have some questions as follow:
1. Did you confirm the two mutations described in other material, hair? Saliva?
2. If the authors want to relate the mutation to clinical aspects, the case must be presented in a complete and exhaustive manner.
3. Did the patient require hospitalization in intensive care?
4. Provide the maximum ferritin value observed in this patient.
5. Why the delay of using cyclosporine and etoposide in this patient?
6. There is a disparity between the text and the Figure 1 A. The graphic does not show the treatment with alemtuzumab for example.
7. Figure 1, A: the graphic mixed events with therapies and it is not enough clear to follow the events in this case. I would suggest improving this graphic.
8. Figure 1, B: the authors should provide reference values for all parameters.
9. Were the mutations search in the family? Please provide the results. Discuss this point.
10. Who was the donor, related, unrelated, identical, etc?
11. Describe post-transplant follow-up, engraftment, initiation of GVHD disease, chimerism. Molecular results regarding the described mutations.
12. It is important for the reader to understand whether the patient was in CR of her Hemophagocytosis syndrome in the post-transplant phase. Please provide the information.
13. Table 1 belongs to review of the literature; it should not be referred in the case description. Please relocate it in the manuscript.
14. Supplementary table 1 was not available for the review.
Author Response
We are grateful to the reviewer for the appreciation of our work and the insightful aspects highlighted. We aim to address each point individually:
Did you confirm the two mutations described in other material, hair? Saliva?
- That is an excellent comment. Confirmation of the mutation was not necessary as we were also able to detect the mutation in the peripheral blood. This means that both alterations are on the germline. The responsible reference laboratory confirmed this. In order to prove the pathogenicity of the R104C alteration, an additive analysis was carried out using the online production program "PolyPhen2". The possible pathogenicity was confirmed. We now also list this in the manuscript (page 5, lines 173 – 175; page 7, lines 235 - 237).
If the authors want to relate the mutation to clinical aspects, the case must be presented in a complete and exhaustive manner.
- We completely agree with the reviewer. We have taken the comment very seriously and have therefore gone into much more detail in the case presentation (page 6, line 203; pages 6 - 7, lines 217 – 227; page 7, lines 230 – 231; page 7, lines 235 – 242; page 7, lines 244 – 246; page 7 - 8, lines 249 – 252; pages 7 – 8, lines 254 – 267; page 8, lines 268 - 279).
Did the patient require hospitalization in intensive care?
- It was only in the final weeks and months of the course of the disease that the patient required high care medical treatment. The patient received medical care in the high care area for a total of 75 days. The first stay in the high care area (71 days) was in the context of the allogeneic stem cell transplantation. This can now also be read in detail in the manuscript (page 8, lines 275 - 276).
Provide the maximum ferritin value observed in this patient.
- The highest measured ferritin value in serum before allogeneic stem cell transplantation is 1032ng/ml and the highest value after allogeneic stem cell transplantation in acute GvHD is 3324ng/ml. The highest measured value of the soluble IL2 receptor before allogeneic stem cell transplantation is 4691U/ml. We now describe the course of the inflammation markers in much more detail in the revised version of the manuscript (pages 6 - 7, lines 217 – 219; Table 1).
Why the delay of using cyclosporine and etoposide in this patient?
- We would like to thank the reviewer for this important comment. The decision to refrain from employing etoposide and ciclosporin in the initial therapeutic intervention was multifaceted. The rationale encompassed the ambiguous constellation of findings arising from the negative degranulation test and the unique genetic compound heterozygosity, not hitherto documented in the scientific literature. Additionally, the patient's age at the onset of primary HLH was considerably advanced. Furthermore, concerns were raised regarding the potential for pronounced ovarian toxicity. The therapeutic approach would have undoubtedly escalated to incorporate etoposide if a robust remission had not been achieved with the initial therapeutic regimen involving IVIGs and prednisolone. We have now integrated this into the discussion of the manuscript (pages 9 -10, lines 325 – 333).
There is a disparity between the text and the Figure 1 A. The graphic does not show the treatment with alemtuzumab for example.
- We thank the reviewer for this helpful advice. Accordingly, we have renewed Figure 1A and evaluated the therapy sequence once again in detail and have now adjusted it (pages 7 - 8, lines 244 – 272).
Figure 1, A: the graphic mixed events with therapies and it is not enough clear to follow the events in this case. I would suggest improving this graphic.
- We agree with the reviewer in every respect. In order to visualize the decisive events of the clinical course and the sequence of therapy in the best possible way, we now use the Swimmer Plot divided into two bars (Figure 1A).
Figure 1, B: the authors should provide reference values for all parameters.
- We are grateful for the suggested addition, which we have gladly included in the revision of the illustration. Missing values were added, where possible and available, after repeated intensive review of the electronic patient file (Table 1).
Were the mutations search in the family? Please provide the results. Discuss this point.
- We agree with the reviewer that this is relevant additional information for this case. Of course, we have recommended human genetic counseling to the family. Due to the dramatic nature of this case and the high frequency of treatment, the human genetic counseling was later than the treatment period we report on here. Unfortunately, we therefore do not know the outcome of this counseling. In the course of the donor search, the patient's sister was considered. A comparable genetic constellation could be ruled out in advance. We were happy to include this point in the discussion of the revised manuscript (page 7 - 8, lines 277 – 279).
- Who was the donor, related, unrelated, identical, etc.?
- Describe post-transplant follow-up, engraftment, initiation of GVHD disease, chimerism. Molecular results regarding the described mutations.
- We thank the reviewer for this question. This was an HLA-identical unrelated donor. We now provide additional information on the transplantation (including conditioning, etc.) in the text (pages 7 - 8, lines 254 – 267).
- In addition, we now provide further details on the course after allogeneic stem cell transplantation, the course of GvHD and molecular remission tests (pages 7 - 8, lines 254 – 267).
It is important for the reader to understand whether the patient was in CR of her Hemophagocytosis syndrome in the post-transplant phase. Please provide the information.
- In the revised version of the manuscript, we have now made sure that the remission data are presented in more detail (page 7, lines 230 – 231; page 7, lines 244 – 246; page 8, lines 264 - 267).
Table 1 belongs to review of the literature; it should not be referred in the case description. Please relocate it in the manuscript.
- We absolutely agree with the reviewer on this point. The reference is now given in the corresponding review of the literature (Table 2, page 8, line 283).
Supplementary table 1 was not available for the review.
We apologize to the reviewers for this inconvenience. We hope that the renewed version of Supplementary Table 1 (excel sheet) is now available to the reviewers.

Round 2
Reviewer 1 Report
Comments and Suggestions for Authors
-
Comments on the Quality of English Language-
Reviewer 2 Report
Comments and Suggestions for Authors
The Authors have addressed all concerns.
Reviewer 3 Report
Comments and Suggestions for Authors
The authors answered my questions, I have no further comments.